# Local and Bayesian Survival FDR Estimations to Identify Reliable Associations in Whole Genome of Bread Wheat

**DOI:** 10.3390/ijms241814011

**Published:** 2023-09-12

**Authors:** Mohammad Bahman Sadeqi, Agim Ballvora, Jens Léon

**Affiliations:** INRES-Plant Breeding, Rheinische Friedrich-Wilhelms-Universität Bonn, 53113 Bonn, Germany; mbsadeghi1@gmail.com (M.B.S.); j.leon@uni-bonn.de (J.L.)

**Keywords:** GWAS, local FDR, Bayesian survival analysis, wheat genome, grain yield

## Abstract

Estimating the FDR significance threshold in genome-wide association studies remains a major challenge in distinguishing true positive hypotheses from false positive and negative errors. Several comparative methods for multiple testing comparison have been developed to determine the significance threshold; however, these methods may be overly conservative and lead to an increase in false negative results. The local FDR approach is suitable for testing many associations simultaneously based on the empirical Bayes perspective. In the local FDR, the maximum likelihood estimator is sensitive to bias when the GWAS model contains two or more explanatory variables as genetic parameters simultaneously. The main criticism of local FDR is that it focuses only locally on the effects of single nucleotide polymorphism (SNP) in tails of distribution, whereas the signal associations are distributed across the whole genome. The advantage of the Bayesian perspective is that knowledge of prior distribution comes from other genetic parameters included in the GWAS model, such as linkage disequilibrium (LD) analysis, minor allele frequency (MAF) and call rate of significant associations. We also proposed Bayesian survival FDR to solve the multi-collinearity and large-scale problems, respectively, in grain yield (GY) vector in bread wheat with large-scale SNP information. The objective of this study was to obtain a short list of SNPs that are reliably associated with GY under low and high levels of nitrogen (N) in the population. The five top significant SNPs were compared with different Bayesian models. Based on the time to events in the Bayesian survival analysis, the differentiation between minor and major alleles within the association panel can be identified.

## 1. Introduction

Bread wheat (*Triticum aestivum* L.) is among one of the major crops for food security worldwide which alone contributes 20% of the protein and calories in our daily diet. Increasing Grain yield (GY) is the main goal of wheat breeding programs. GY is a complex trait controlled by many genes with small effects. Under field conditions, both genetic and environmental factors, and interaction between genotype and environment (G × E) affect GY [1]. However it is observed that the analysis of GY related parameters presents challenge. Because the observations should be independent and identical and the residuals must follow normal distribution. In practice, they are affected by genetic and environmental factors and the distribution of data is far from normal [2]. There is only one mean vector obtained from the observations, which is not sufficiently informative to confirm whether the distribution is normal or not. One of the most popular tools to compute true confidence interval for the mean and standard error is bootstrap [3], which is a resampling statistical device to measure the accuracy of observed bias and variance [4] in complex traits like GY. In field experiments, genotypes have random effects, which lead to some outliers among the observations. The best linear unbiased predictors (BLUPs) is utilized in phenotypic mixed linear models to estimate random effects. BLUPs with frequentist perspective can be an analogy to Bayesian inference in dealing with outliers [5]. In parallel, genetic estimated breeding value (GEBV) describes the individual genetic merit to produce superior offspring, and correlation between GEBVs and BLUPs is a common approach to check the quality of GY vector [6]. GEBVs of complex traits can be computed by BLUPs via genome wide association study (GWAS) models. So, GWAS is a successful and fast strategy for genetic dissection of complex traits in plants [7]. The objective of GWAS is to determine the presence of a significant relationship between genotypes and traits of interest. Therefore, understanding variations in rare phenotypes among complex traits and associations between large number of markers (typically SNPs) and a given trait are of great interest [8]. It is particularly useful when little information on the genetic control of a quantitative complex trait is available. The main challenge is to find significant reliable associations in GWAS results. Population structure and genomic relationship (kinship) matrix are two main components involved in the GWAS model that can reduce structural and systematic effects. Furthermore, there are some genetic hyper-parameters such as marker effect, minor allele frequency (MAF), number of call rate for each marker [9] and epistasis effects between loci [10], which are not presented in usual GWAS models, but can improve the power and accuracy of models significantly. Moreover, various GWAS models, such as single locus association, multi-locus association [11,12], Bayesian whole genome regression and whole genome variance, have potential to attain reliable significant associations. In the single locus association model, population structure and environmental factors are taken as fixed effects while phenotypic values and markers are considered as random effects [13]. It further requires Bonferroni correction with false discovery rate (FDR) of 0.05 for pairwise tests between the complex trait and SNPs. At this threshold level, although the risk of a false positive has been reduced, the chance of producing true positive associations is very low. As such, this model has led to many pseudo associations in the results of some studies [14]. Moreover, high over or low under-fitting in single locus association model due to high-density SNPs makes optimization difficult. Meanwhile, the multi-locus association model based on the restricted maximum likelihood (REML) method applies better FDR correction to reduce selection criterion, but due to pairwise comparisons, collinearity in this model remains high. It has also led to over- or under-fitting in the results of GWAS model [15]. Particularly, when the epistasis effects between loci are high, due to low bias and high variance in the model parameters, many pseudo associations arise in the results [16,17,18]. The whole genome variance model provides the analysis of genetic variance based on whole genome regression [19]. The probability of receiving reliable significant associations in this model is still low, due to multicollinearity and heteroscedasticity in model components. The single, multi-locus and whole genome variance models are based on linear regression algorithm, which has low efficiency for big genomic datasets with large-*p-value*-small effects. However, the Bayesian whole genome regression model based on prior and posterior probability distributions is a convenient algorithm to deal with these difficulties [20]. Therefore, the model’s accuracy can be higher than three previous frequentist models, because the complex trait and SNPs are taking prior distribution [21]. In this model, SNP effects and dimensions of genomic dataset are interpreted with Bayes factor (BF), which is easier and more reliable against SNP *p-values* [22]. BF is a summary measure that provides an alternative to the *p-value* for ranking significant associations [23]. In addition, in this model, false coverage rate (FCR) is considered analogous to FDR [24]. FCR covers more dimensions among the selected intervals in the genomic file [25]. Nevertheless, determining the actual prior distribution for the GY vector and SNPs represents a difficult part of computation [26]. Genomic prediction (GP) as a promising technique in molecular breeding provides the possibility to consider the performance of GWAS models. Bayesian survival FDR analysis is a robust strategy based on probability theory to determine the true prior distribution in large-scale genomic datasets [17,27]. Moreover, this analysis is a regularization approach that can be applied as a GP model with high performance [28]. It attempts to estimate all genomic effects among GY vector, while the pseudo effects of covariates are reduced to zero within SNPs [29].

## 2. Local and Bayesian Survival FDR Analysis in GWAS

In the last decade, GWAS has been made feasible by high-throughput genomic technologies based on next-generation sequencing (NGS) techniques. When multiple pairwise association tests are performed between case and control individuals, it becomes difficult to minimize type I error (false positive error) in hypothesis testing simultaneously. Some efforts to control the family wise error rate (FWER) versus multiple pairwise errors were proposed by Bonferroni, then Benjamini [30], who proposed FDR control as an alternative approach to overall type I error. GWAS is very sensitive in identifying the genetic basis of complex traits such as GY with many agronomic components with small effects. The reasons for this sensitivity could be the many genetic variations with small effects among individuals and the variation in genetic structure of sub population. Thus, the power of common GWAS models, such as genome regression models, to control for the FDR threshold is very low to detect non null hypotheses for associations. Efron and Tibshirani developed an extension of FDR for large-scale simultaneous hypothesis testing, called local FDR based on MLE in scale of z-value versus *p-value* [31]. Large-scale genetic association studies are conducted in the hope of discovering signal SNPs involved in the association of complex traits. Identifying the correct FDR to decide between signal associations is the critical key of GWAS. Based on frequent perspective, the final extension of the correct threshold in GWAS is the local FDR approach proposed by Efron [32]. The local FDR approach is suitable for simultaneously testing many associations, based on the empirical Bayes perspective.

In the local FDR framework, the FDR is a measurement of posterior signal SNPs with rule:$$P(z_{i}\left(\pi_{0}f_{0}+\pi_{1}f_{1}+\cdots +\pi_{n}f_{n}\right)>\left(1-\alpha \right)$$
where $z_{i}$ is a t-statistic comparing pairwise SNP associations with $N\left(0,1\right)$ distribution under null hypothesis, π is proportion of true null hypothesis, f the mixture density estimate at midpoint of bin, calculated based on empirical Bayes rule of $z_{i}$ with degree of freedom (df) for fitting the MLE of $z_{i}$, α is significant level of association test. The local FDR (*locFDR*) is then defined as:$$locFDR\left(z_{i}\right)=\frac{z_{i}\left(\pi_{0}f_{0}\right)}{\sum z_{i}\left(\pi_{i}f_{i}\right)}$$

Thus in *locFDR*, maximum likelihood estimator is sensitive to bias when the GWAS model includes two or more explanatory variables as genetic parameters. The results are based on maximum likelihood of marker effects, which is only a point estimation and there is no further information about prior distribution of the signal and followers of the signal associations.

However, the advantage of the Bayesian perspective is that the knowledge of prior distribution comes from other genetic parameters included in the GWAS model, such as linkage disequilibrium (LD) analysis, MAF and call rate of significant associations. In the current study, we propose to solve the multicollinearity and large-scale problems in the GY vector as a complex trait in bread wheat and large-scale SNP information with Bayesian survival analysis. The aim of this study is to obtain a short list of SNPs that are reliably associated to GY vector among three levels of N. The hypothesis to test, for each SNP = SNP_1_,…, SNP_p_ when pairwise comparisons between censored and relapsed observations, are conditionally independent, $x_{SNP}\sim P\left(\theta_{SNP}\right)$, so $x_{SNP}$ is GY values for each genotype. We suppose the SNP effect as parameter space of $\theta_{SNP}\in \Theta$, where Θ is an unobserved scalar parameter that can be partitioned in two parts ($\Theta_{0},\;\Theta_{1}$) such that:

H_0_: $\theta_{SNP}\in \Theta_{0}$ or SNP_i_ is not significantly associated with GY among population.

H_a_: $\theta_{SNP}\in \Theta_{1}$ or SNP_i_ is significantly associated with GY among population.

In Bayesian survival framework, the FDR is a measurement of posterior signal SNPs with rule:$$lP(\theta_{SNP}\in \Theta_{1}\;|x_{SNP})>\left(1-\alpha \right)$$

Where α takes MAF of SNP in the pairwise comparisons of censored and returned observations.

The Bayesian survival FDR (bsFDR) is then defined:$$bsFDR\left(\lambda_{SNP}\right)=\frac{\sum_{SNP}P(\theta_{SNP}\in \Theta_{1}\;|\lambda_{SNP}.\theta_{SNP})}{\sum_{SNP}(\lambda_{SNP}.\;\theta_{SNP})}$$
where $\lambda_{SNP}$ is Bayes survived factor that indicate SNP effect based on MAF value for GY of each genotype.

Estimating a significant FDR threshold in GWAS is still a major challenge to distinguish true positive hypotheses from false positives and negative errors. Several comparative methods for multiple testing have been developed to determine the significance threshold; however, these methods may be overly conservative and lead to an increase in false negative errors. Here, we developed two empirical methods to determine the statistical significance threshold based on SNP- heritability of GY as a target trait. To test the locFDR and bsFDR methods for significance threshold under different distinguished GWAS models, we used the mean of results from three years of field experiments on GY vector in bread wheat under low N (LN) and high N (HN) applications.

## 3. Results

### 3.1. Grain Yield Quality Control

The GY observations obtained from 221 bread wheat genotypes under HN treatment were averaged across three years with a mean value of 6400 and standard division of 145.16 (gr/m^2^). In the histogram plot, the Shapiro *p-value* = 0.0891, which indicates the GY vector follows a normal distribution. However, we considered this test inadequate, so to focus more on the quality of observations, the random vectors were repeated 2000 times with replacement from original GY vector was generated and Bayesian bootstrap *p-value* = 0.00325, which shows the Shapiro *p-value* is accurate (Figure 1b). Consequently, a confidence interval of 95% and standard error in simulated samples are close to original observations. Correlation between BLUPs and BLUEs as analog for EBVs is high (*r* = 0.814), showing that the effects of genotypes are distributed randomly and the quality of GY vector is acceptable to use in GWAS and GP models. Moreover, under LN treatment after removing outliers the Bayesian bootstrap *p-value* = 0.3048, which shows the Shapiro *p-value* = 0.2462 follows a normal distribution and is ready to apply in GWAS models (Figure 1a).

### 3.2. Population Structure Analysis

In Figure 2, based on the population of 221 genotypes, genetic data are divided into three clusters which implies these genotypes differ from each other in genetic content.

The variance is explained by PCA being normally distributed in ten PCs and the first three PCs with explained variance 20.5, 10.8 and 9.7%, respectively, are considerable as PC number in the GWAS analysis. The allocation of genotypes in related clusters could be attributed to their genetic descent and common ancestry. The *F_st_* plot based on 150 K SNPs with MAF greater than 0.05, indicates how heterozygosity varies in sub-populations in comparison with the whole population. It is thus considered a threshold of calculated proportion of homozygosity in whole genome, which is clearly observed in Figure 2d and the proportion of homozygosity in the population is acceptable.

### 3.3. GWAS and GP Model Selection

To determine the best GWAS model, local FDR analysis on SNP *p-values* for each model was run.

In Figure 3a, the results under low N show that in the variance component model (*sommer*) the standard error is minimum (sigma = 0.048), but Bayesian whole genome regression (*NAM*) exhibits the highest Delta (μ) = 0.589 and highest Proportion (H_0_) = 0.625, which shows that the false positive rate to reject H_0_ via *sommer* under LN treatment is not high, so this minimum sigma = 0.048 is not reliable. For all GWAS models, the local FDR plot was revealed in one tail.

In Figure 3b, the results under high N show that for Bayesian whole genome regression (*NAM*) the standard error is minimum (sigma = 0.062), the Delta is highest (μ) = 0.661 and Proportion is highest (H_0_) = 1.034, in front in variance component model, sigma = 0.664 is highest and the higher sigma with lower Delta (μ) = 0.358, and lower Proportion (H_0_) = 0.433, representing lower model accuracy, so the *sommer* model was removed from further analysis.

For *rrBLUP* and *NAM*, the local FDR plot was one tail but in *mlmm* model the two tails of significant SNPs was revealed. To check the performance of the remaining three GWAS models, the Bayesian survival analysis based on highest significant association was performed for GY vector under LN and HN treatments separately, and the mean and standard error of survived SNPs (*S(SNP_i_*)) were calculated. To measure the accuracy of each model, Akaike information criterion (AIC) and Bayesian information criterion (BIC) were calculated and the results are shown in Figure 4. Under LN, GY vector in the *rrBLUP* and *mlmm* models was the only major allele that survived with mean of *S*(*SNP_i_*) 0.547 and 0.481. However, in the NAM model both major and minor alleles during survival analysis have remained. Moreover, the NAM displayed the minimum standard error (SE *S*(*SNP_i_*) = 0.189) in comparison to the other two models. The AIC and BIC were 150,408 and 150,867, respectively, which are lower than *rrBLUP* and *mlmm*. For GY vector under HN condition, the bootstrapping method was applied and in the *mlmm* model only the major allele survived with mean of *S*(*SNP_i_*) 0.597. However, in the *rrBLUP* both major and minor alleles and in the NAM all major, minor, heterozygous and missing (NA) alleles during survival analysis remained. In addition, the NAM exhibited minimum SE *S*(*SNP_i_*) = 0.104 in comparison to the other two models. Here, the AIC and BIC values were also less than those against *rrBLUP* and *mlmm*. Therefore, both local FDR and Bayesian survival analyses have confirmed NAM to be the best GWAS model with minimum residual errors on the SNPs. The Manhattan and QQ-plots for NAM model for GY vector under HN are shown in Figure 5a,b.

The association points well above the gray area in QQ-plot correspond to GWAS hits and in Manhattan plot by threshold *−log10* (*p* = 4), the continuous points on chromosome region 3A specify the reliable SNPs, which show association with investigated trait (Table 1 and Table 2).

### 3.4. SNP Effect Estimation

In this study, we were also interested in examining the impact of prior distribution on the SNP *p-values* of NAM as the best model of GWAS and estimation accuracy. Therefore, the GP models were assessed with six different Bayesian prior distributions: A, B, C, LASSO, ridge regression and survival. For all prior distributions, the markers with null effects were removed from analysis. For Bayesian survival prior distribution, $\lambda_{0}=1$ and $\exp\left(\beta_{GY}\right)=0.001$, was assigned to obtain flat prior. To calculate $\lambda \left(SNP_{i}\right)$ and *p-values*, a Markov chain Monte Carlo (MCMC) of length 150,000 iterations was considered after a burning bootstrap period of 2000 iterations. The *p-values* generated from all prior distributions were converted to z-values for easier interpretation and minimizing the effect of the SNP dimension. By comparing the box plots of the SNP Z-scores generated from six different Bayesian prior distributions (Figure 6a), the Bayesian LASSO and Bayesian survival models exhibit highest accuracy by correlation between predicted genetic values ($\hat{g}$) with true breeding values (g), $r_{Bayes_{L}}\left(\hat{g},g\right)$ = 0.8722 and $r_{Bayes_{Surv}}\left(\hat{g},g\right)$ = 0.8876. The top five significant SNPs received from NAM model (Figure 6a) were compared with different Bayesian models. Only the Bayesian survival analysis can identify the differentials within them due to time to events among minor and major alleles within association panel. With this approach, the first two signal SNPs show a trend, and share reliable associations. However, for other Bayesian platforms there is no trend among SNP effects (Figure 6b).

## 4. Discussion

Nitrogen is the main nutrient for canopy growth and photosynthesis, which is responsible for GY and quality. Allelic variation for GY under low and high N could be high due to large mutations in the signal SNPs within candidate genes. Moreover, considerable effort is required to identify all involved variants among complex traits. Therefore, the main challenge in GWAS is to find significant and reliable associations related to a given complex trait. To address this dilemma, local FDR correction and Bayesian survival analysis, two precise and efficient computational approaches, serve as different filters to determine the best GWAS and GP models and consequently obtain a reliable association in the result of the best selected model. Mixed populations and variants with small effect sizes or rare alleles in kinship coefficients derived from marker information or outliers in the phenotypic vector, may lead to causal signals and false association between marker and complex trait. In parallel with these genetic parameters, there are some genetic hyper-parameters such as panel size, number of markers, MAF and number of call rates for each marker, that are not represented in the given model, but which have effects on the performance of the model and accuracy of results. In the GWAS and GP, one side of the model is allocated to the complex trait. To reduce the multicollinearity and heteroscedasticity in phenotypic observations, a tradeoff between bias and variance is required. Outliers in the GY vector are the first cause of bias results in GWAS. We used bootstrapping as a powerful and well-known tool to deal with outliers in the GY vector. However, in this parametric resampling method, the GY vector is simulated to obtain the confidence interval of the mean and standard error of the phenotypic observations. However, randomization with global replacement in parametric bootstrapping leads to underestimation of the variation of the complex trait. In this study, in addition to dealing with outliers with usual procedures, we applied Bayesian bootstrap using MCMC state resampling, which can be a good alternative to control underestimation of variance. Because the observations appeared with a certain probability with bias and variance tradeoff on the results. This probability based on prior distribution offers more precise estimation. In the results, Bayesian bootstrap *p-value* shows more significance with lower sampling error among the GY vector (Figure 1a). Therefore, the correlation between BLUPs and BLUEs is higher than 0.8 among prior distribution of the GY vector. However, the low replicability and reliability of results in the linear GWAS models, such as single-locus, multi-locus association and whole-genome variance models, pose a major challenge. Basically, in self-pollinated wheat crops, there are some non-additive relationships between loci, such as codominant and epistasis, due to heterogeneity within the locus. However, linear GWAS models do not account for this problem, and the genomic relationship matrix is estimated based only on additive fixed effects between SNPs. Consequently, the false positive rate and type I error are high, and many SNPs with true non-zero effects behave like null effects and, conversely, many SNPs with null effects show significant associations in the results. In contrast, in Bayesian whole genome regression, SNPs are considered as random effects, which introduces MAF as the main genetic hyper-parameter in the model. Additionally, the effects of minor and major alleles are accounted for via the $XZ_{\alpha \beta }$ component in the model, which reveals the allelic effects from the interaction between population structure and kinship matrix. All these different types of information, including genetic parameters and hyper-parameters, bear implications for assessing reliable associations in the GWAS results. With the Bayesian framework, it is possible to include all this information in both sides of the model, which greatly reduces the type I and II errors due to the posterior specifications for each component of the model. In the present work, we have studied two precise and efficient computational approaches as different filters to determine the best GWAS and GP models and consequently obtain reliable association in the results of the best selected model.

Local FDR as frequentist inference with Fisher information background was applied to check the distribution of SNPs especially in the tails as critical regions of the distribution. This method, based on maximum likelihood estimation, is more precise for determining the effect size and strength of a large-scale genomic file. As can be observed in the results, the expected mean likelihoods based on SNP information are acceptable, but the standard error in the models with a scale of *−log10* (*SNP*) is still high. The main criticism of this approach is that it locally focuses only on SNP effects in tails, whereas signaling associations were distributed throughout the genome. Moreover, the magnitude of SNP effects alone may be insufficient to inform a conclusion about the performance of the model. For example, the degree of freedom (df) to fit the estimated density among SNPs is affected by the MAF and the quality of SNP chip (e.g., NAs and imputation). A larger df is required for sharper tails in $f\left(zz\right)$, but in the *mlmm* model df is low, and consequently the sigma is partially low, which may lead to underestimation or overestimation of the model. With this limitation, we also considered whether we should use this approach because the proportion of false rejection hypotheses is larger in this approach than in the family wise error rate (FWER) approach. The most interesting result of this approach is that in the variance component GWAS model, proportion of H_0_ (sigma/mean) = 1.034 is the highest due to the highest sigma (SNP standard error), thus this model can clearly be removed due to its very low power. In the Bayesian whole genome regression with a low proportion of H_0_ (sigma/mean) = 0.633, the putative signal SNPs in the tails were very well covered and the model produced the lowest error (sigma = 0.062).

Bayesian survival FDR analysis as a modern inference based on posterior estimation of SNP effects is commonly used for large-scale genomic data with small effects. In the survival part with the $(-\Delta \left(SNP_{i=0}^{n}\right)$ component of the approach, we included the minor and major alleles in each SNP based on the time to events, which is necessary to achieve the expected prior distribution. Therefore, the MAF and the quality of SNP chip were considered in the computation. In GWAS with the original SNP information among GY vector, only in the *NAM* model were both minor and major alleles revealed in the results with the highest AIC and BIC criteria versus two models. In GWAS with SNP simulation, all allelic effects including major, minor, heterozygous and missing were revealed in the same model, which may present good evidence supporting this model as the best performing. Thus, we found that this is in good agreement with [29] to minimize the type I error at the significance level of MAF. As shown in Figure 5 and Figure 6b, the relationship between effect size and MAF is not strong for SNPs with higher MAF value. The difference is claimed for SNPs with low MAFs (from SNP3: AX-110385692 to SNP5: BS00094057_51). In this case, it indicates that the informative nature of $locFDR\left(z_{i}\right)$ is lower than $bsFDR\left(\lambda_{SNP}\right)$ to determine reliable associations, which closely aligns with the results of [23,33]. Binary MCMC sampling is a renewable technique in survival Bayesian inference to generate large-scale genomic datasets; however, it is affected by long computation times in the linear algorithms. In the Bayesian part, we solved this technical difficulty using the exponential form $\exp(-\Delta \left(SNP_{i=0}^{n}\right)$ to predict the SNP prior distribution. For both the local FDR and Bayesian survival approaches, the standard error of the *NAM* model was lower than the others. To compare the performance of the Bayesian survival model with other commonly used Bayesian platforms, we proposed $\lambda \left(SNP_{i}\right)$ based on covariate effects between SNPs. These covariate effects have negative implications for the expected prior distribution in all Bayesian approaches, including survival. To solve this problem, $\beta_{GY}$ should take an exponential form based on $\lambda_{0}$ which allows the GP algorithms to examine SNP effects with minimal bias and variance. To compare the accuracy of GP models, we used Z-score because it increases the mean and variance of genetic value for each genotype. However, it makes interpretation easier and more accurate than *p-value* and Bayes factor with a smaller comparison interval. The five top significant SNPs were compared with different Bayesian models. Only Bayesian survival analysis can identify the difference between minor and major alleles based on time to events. In $bsFDR\left(\lambda_{SNP}\right),\;\;\lambda_{SNP}$ is a semi-parametric statistic because it is based on MAF and allelic scores of genotypes simultaneously [34]. It seems that the semi-parametric empirical Bayes factor better controls both false positive and false negative errors in GWAS to identify allelic variations and find templates for significant SNPs. Therefore, we propose the utilization of different GWAS models at more N levels to increase the replicability of the posterior probability [35] of the identified signal associations.

## 5. Materials and Methods

### 5.1. Plant Materials and Field Experiment

In this study, a set of 221 bread wheat genotypes from Breeding Innovations In Wheat for resilient Cropping Systems (BRIWECS) project were cultivated as split-plots design, in three cropping seasons 2018, 2019 and 2020 at the agricultural research station Campus Klein-Altendorf, University of Bonn, Germany. We recorded the GY value of each genotype under high and low N fertilizer annually. Then, the average GY values of the genotypes were used as GY vector for subsequent analysis. The outliers in the GY vector were checked. To deal with outliers, they were kept since they reflected the actual field values across all years. The distribution of the residuals was checked using Shapiro normality test. To improve the quality of the vector, 2000 repeated random samples with replacement were generated from the original GY vector using R/*bootstrap*, and then Bayesian bootstrap *p-value* was calculated. To also control genotype random effect, the EBVs and BLUPs were based on broad sense heritability [36] plotted.

### 5.2. SNP Quality Control

A platform of 150 K affymetrix SNP Chip was used to apply the GY vector in the GWAS and GP models. The SNPs with MAF ≤ 0.05 were removed due to monomorphism in the marker. After checking SNPs that deviated from the Hardy–Weinberg equilibrium (HWE), only 22,489 polymorphic SNP markers remained, which were used in GWAS and GP analyses.

### 5.3. Population Structure

Cumulative variance explained by the eigenvalues of the principal components was calculated, and then discriminant analysis of principal components (DAPC), which minimizes genetic variation within clusters, was also conducted using R/*adegenet* for assessing the population structure. To determine the actual number of PCs retained in the DAPC, cross validation (CV) was performed to simulate the low and high numbers of PCs in the model. To control outliers in the population clusters, information from the identical-by-state (IBS), which measures differences in allelic states, and the fixation index ($F_{st}),$ which refers to the amount of heterozygosity at different levels of population structure, were calculated based on covariance estimation [37]. To detect the regions that might be involved in the linkage pattern of the population, the whole genome wide plot of $F_{st}$ was constructed.

### 5.4. Construction of Genomic Relationship Matrix (GRM)

Basically, the GRM as kinship matrix is used in GWAS and GP models. Based on the method suggested by (Mathew et al. [38]), the covariance between individuals $g_{i}$ and $g_{j}$ can be equal by covariance of $SNP_{ij}$, therefore the GRM was calculated using $G=\frac{\Sigma_{k=1}^{L}{\left(g_{ik}-p_{k}\right)}^{2}}{\Sigma_{k=1}^{L}p_{k}\left(1-p_{k}\right)}$, and $p_{k}=\frac{1}{n}\Sigma_{i=1}^{n}g_{ik}$ where, L is number of loci, $p_{k}$ is MAF for the locus k and $g_{i}$.

### 5.5. GWAS and GP Models

A total of four different GWAS models, including single-locus association with R/*rrBLUP* [39], multi-locus association with R/*mlmm.gwas* [11], variance component association with R/*sommer* [40] and Bayesian whole genome regression with R/*NAM* [41] at low and high N levels were performed to detect the association between SNPs and GY as a target trait. The single-locus association model was fitted in the adjusted form the mixed linear model as y=Xβ+Zu+e, where y is the trait vector, X is the fixed effects matrix, β is the vector of coefficients including principal components and population structure, Z the matrix of random SNP effects coded as (−1, 0 and 1), $V\left(u\right)=K\sigma_{g}^{2}$, where K is the GRM as kinship matrix and $\sigma_{g}^{2}$ is additive genetic variance with IBS basis. It was removed from the model due to convergence of N × Y to zero. Multi-locus association model, form $y_{i=1}^{n}=\mu +\overset{m}{\underset{j=1}{\sum }}M_{.j}\beta_{j}+e$, where $y_{i=1}^{n}$ is trait vector with n genotypes, m is total number of SNPs, $M_{ij}$ is the matrix of random SNP effects coded as (0, 1 and 2) and $\beta_{j}$ is the vector of SNP effects and $H_{0}$ is given in terms of $\beta =\sigma_{g}^{2}=0$. Once the *−log* (*p-value*) is above the FDR threshold line, $H_{0}$ is rejected. In the variance component model, $y=\overset{c}{\underset{i=1}{\sum }}K_{i}\sigma_{g}^{2}+e$, where c is non-overlapping classes of SNPs, $K_{i}$ is the class of kinship matrix based on genomic data and the components of genetic variance are from REML of the mean of SNP information. Bayesian whole genome regression model was fitted to $y=\mu +X\alpha +Z\beta +XZ_{\alpha \beta }+e$, where y is trait vector, X is matrix of genotypes and SNPs, α is corresponding vector of SNP effects that captures small effects of all SNPs, β is vector that captures additional effects of SNPs with large effect based on Bayes factor.

GP models were performed with Bayesian whole genome regression based on the following model: $y=\mu +X\beta +\left(Z_{1}Z_{2}\right)u+e$, where, y is the GY vector, X, is SNP information for the genotypes, β is regression coefficient for each SNP, indicating the SNP effect in the model, $Z_{1}$ is effect of major allele, $Z_{2}$ is effect of minor allele, u is vector of Bayes factor for SNP matrix and e is implies to residual term [42]. Bayesian models including Bayes A, B, C, LASSO, RR and Bayesian survival analysis, were then applied to predict genomic estimated breeding values (GEBVs). The CV approach was used to evaluate the accuracy of each model. To determine the best model, correlations $\rho \left(\overset{ˇ}{y}EBV,\;\overset{ˇ}{y}GEBV\right)$ were estimated for each GP model.

### 5.6. SNP Effect Estimation under locFDR and bsFDR

To check the distribution of SNPs *p-value*, for each GWAS model under low and high N levels, zz vector was created under the null hypothesis *N* (0, 1), which is necessary, when SNPs *p-value* is very far from normal distribution. To fit the density of *f* (*zz*) with heavy tails, degree of freedom (*df*) was determined based on SNP sample size. Then, empirical null hypotheses were used to estimate the parameters of *f* (*zz*) by maximum likelihood, indicating the accuracy of each GWAS model, and the results were visualized using R/*locfdr* [32]. Based on $locFDR\left(z_{i}\right)$ function, the significant associations above FDR threshold line of 0.05 were selected for the GWAS models. The survival function was represented as a cumulative hazard function using R/*survival* [43],$\;S\left(SNP_{i}\right)=\exp\left(-\Delta \left(SNP_{i=0}^{n}\right)\right)$, where, $S\left(snp\right)$ is survived coefficient for SNP, which estimates the tendency between uniformity in the bottom of SNPs *p-value* distribution and the peaks, while there were no other observations in the bottom of distribution $S\left(y_{SNP}\right)$. Based on the MAF and missing values of each SNP, the prior distribution of significant associations were calculated using the function $bsFDR\left(\lambda_{SNP}\right)$, for the GWAS models. To check for covariate effects within SNPs in the GP models, the Bayesian survival function was applied in the form of $\lambda \left(SNP_{i}\right)=\lambda_{0}\left(SNP\right)\exp\left(\beta_{GY}\right)$, where, $\lambda \left(SNP_{i}\right)$ is the Bayes survived factor indicating the SNP effect, $\lambda_{0}$ is the baseline for function and $\beta_{GY}$ is regression coefficient of the whole genome under the GY vector.

## Figures and Tables

**Figure 1 ijms-24-14011-f001:**
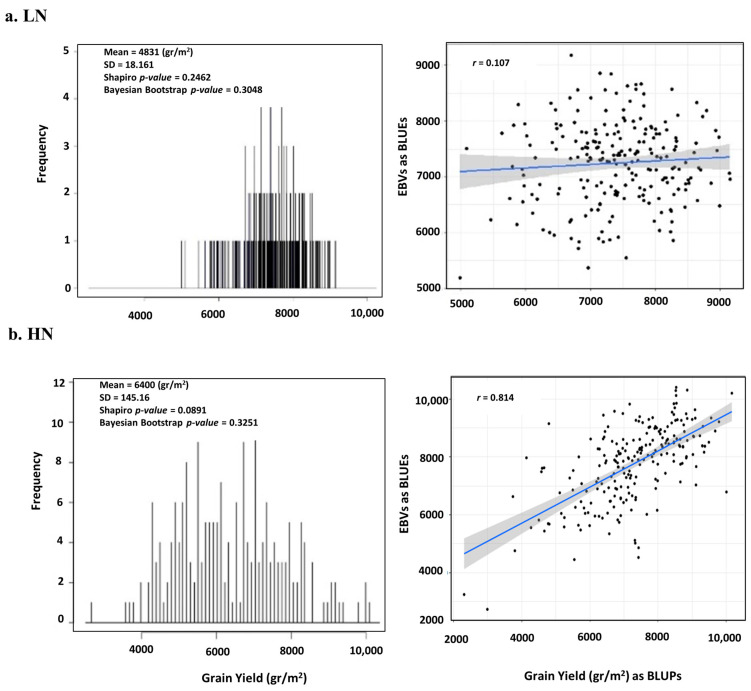
(**a**) Left, GY vector under LN treatment after removing outliers the Bayesian bootstrap *p-value* = 0.3048, which shows the Shapiro *p-value* = 0.2462 is following normal distribution, but on the right, the correlation between BLUPs and BLUEs as analog for EBVs is low (r = 0.107). (**b**) Left, GY vector under HN treatment, mean of three agronomic years 2018, 2019 and 2020, quality control via Shapiro *p-value* and Bayesian bootstrap *p-value*, both *p*-values represent acceptable management of outliers in the vector; right, the correlation between BLUPs and BLUEs as analog for EBVs is high (r = 0.814).

**Figure 2 ijms-24-14011-f002:**
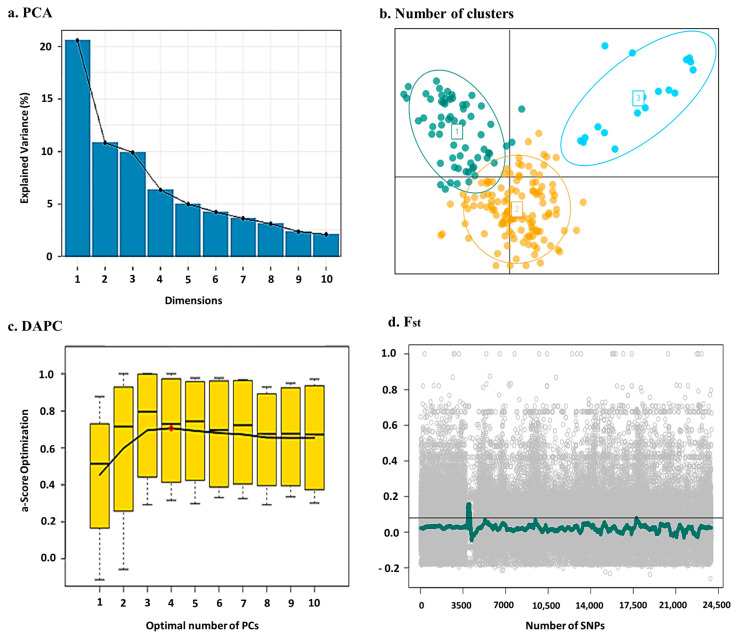
(**a**,**b**) PCA-explained variance and number of clusters by PCs based on 150 K SNP Chip on 221 bread wheat genotypes, (**c**) discriminate analysis of principle components with alpha-score based on genetic diversity between clusters, (**d**) fixation index using 150 K SNP Chips of the genetic variation among and within population.

**Figure 3 ijms-24-14011-f003:**
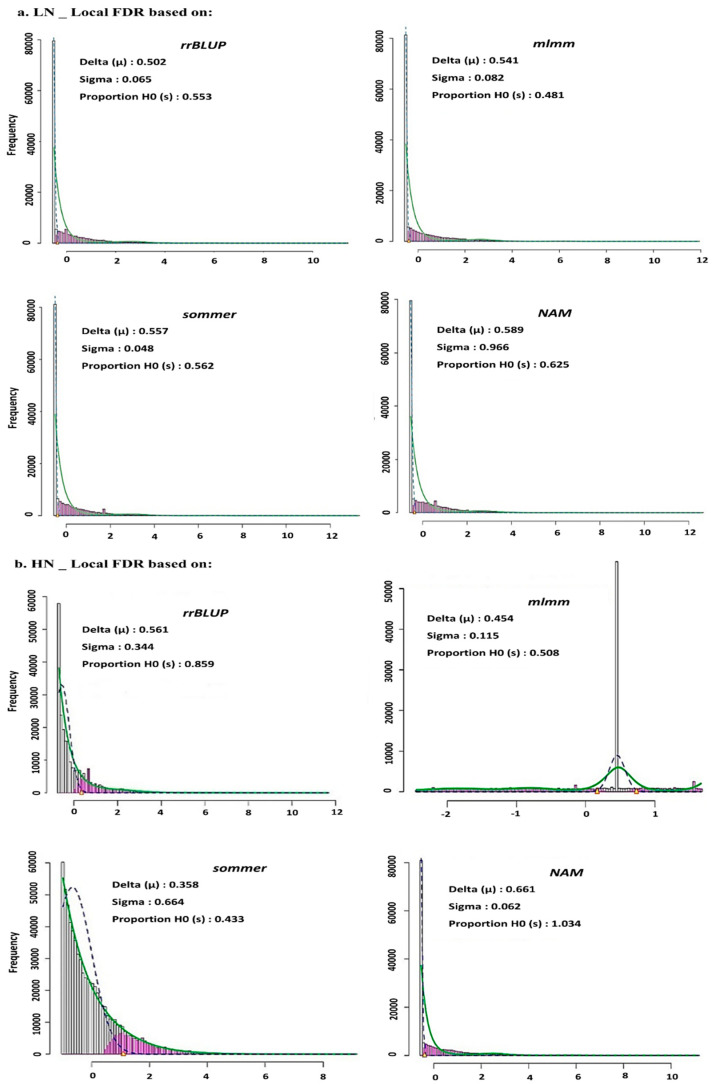
(**a**) GWAS model selection based on local FDR approach, in each model: Delta shows maximum likelihood for mean estimation, Sigma is standard error and Proportion of H_0_ refers to proportion of false rejection hypotheses. All three estimators are important to determine the accuracy and performance of GWAS model for grain yield under low N fertilizer. (**b**) GWAS model selection based on local FDR approach, in each model: Delta shows maximum likelihood for mean estimation, Sigma is standard error and Proportion of H_0_ refers proportion of false rejection hypotheses. All three estimators are important to determine the accuracy and performance of GWAS model for grain yield under high N fertilizer.

**Figure 4 ijms-24-14011-f004:**
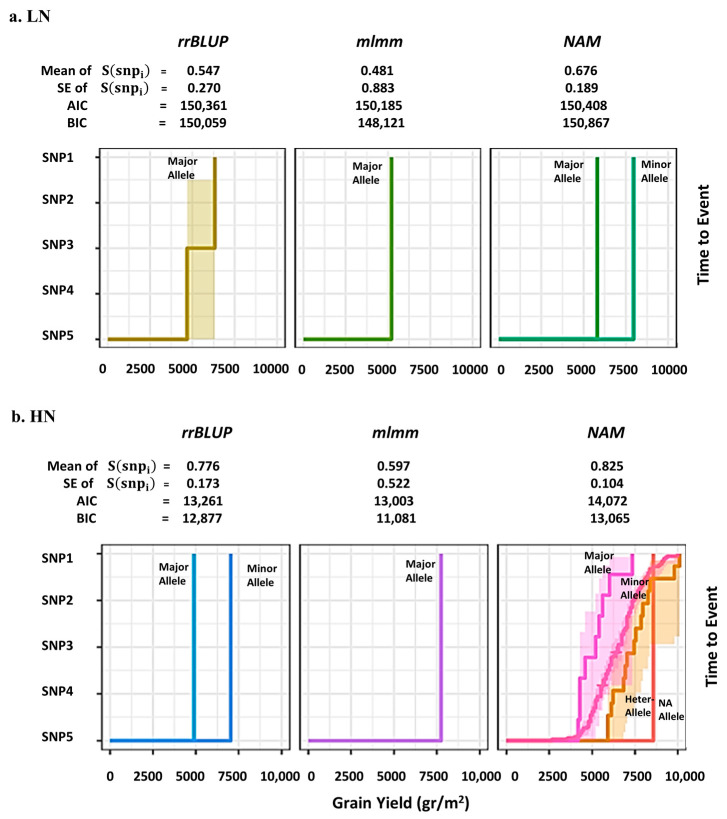
GWAS model selection based on Bayesian survival analysis, the minor and major alleles in each SNP were involved in computation based on time to events, which is necessary to attain expected SNP prior distribution. SNP1: AX-158576714, SNP2: AX-109898892, SNP3: AX-110385692, SNP4: AX-109470057 and SNP5: BS00094057_51. (**a**) Under LN, GY vector in the *rrBLUP* and *mlmm* models only major allele survived with mean of *S*(*SNP_i_*) 0.547 and 0.481, but in the NAM model both major and minor alleles during survival analysis remained. (**b**) Under HN condition, in the *mlmm* model only the major allele survived with mean of *S*(*SNP_i_*) 0.597, but in the *rrBLUP* both major and minor alleles and in the NAM all major, minor, heterozygous and missing (NA) alleles during survival analysis remained.

**Figure 5 ijms-24-14011-f005:**
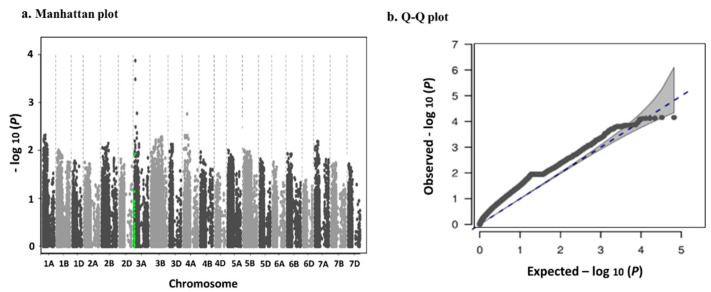
(**a**) Manhattan upon Bayesian whole genome regression GWAS model, the top five significant SNPs include Ax-110385692 in chr. 3A, Ax-158547970 in chr. 3A, Ax-158538619 in chr. 4A, Ax-158522989 in chr. 3A and Ax-109470057 in chr. 3A, were plotted with NAM package. (**b**) In QQ plot, all top five significant SNPs closely follow the Chi-square association line trend.

**Figure 6 ijms-24-14011-f006:**
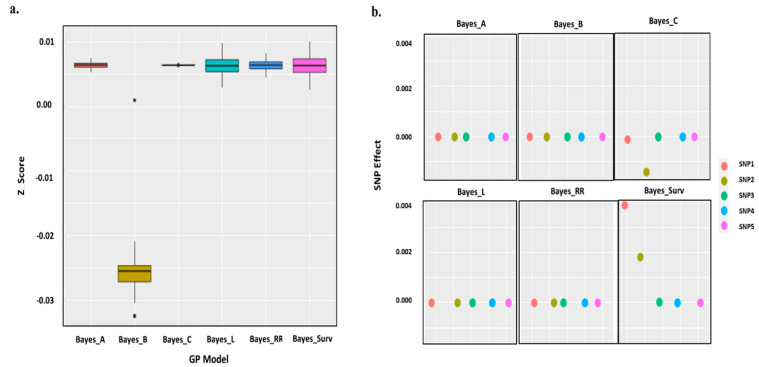
(**a**) Accuracy of GP models based on Bayesian inference, Z-score was used due to increase mean deviation in genetic value for each genotype. (**b**) The top five significant SNPs were compared with different Bayesian models. Only the Bayesian survival analysis can identify the differentials within them due to time to events among minor and major alleles within association panel. SNP1: AX-158576714, SNP2: AX-109898892, SNP3: AX-110385692, SNP4: AX-109470057 and SNP5: BS00094057_51.

**Table 1 ijms-24-14011-t001:** GWAS results using rrBLUP, mlmm, Sommer and NAM models for GY vector of 221 wheat genotypes under low N treatment.

GWAS Model	N Level	SNP	Chr ^a^	Pos ^b^	Maj. Allele ^c^	MAF ^d^	−log(*p-Value*) ^e^	ML $\left(locFDR\left(z_{i}\right)\right)$ ^f^	P $(bsFDR\left(\lambda_{SNP}\right)$) ^g^
*rrBLUP*	LN	AX-158596644	2A	136,652,249	NA	---	4.456407631	0.502	0.195
	LN	AX-158568302	7D	55,688,624	NA	---	4.256507384		0.179
	LN	AX-109865927	6D	160,078,273	AA	0.176	3.669168314		0.235
	LN	AX-158576714	3A	69,099,202	CC	0.181	3.288885		0.237
	LN	AX-109898892	3A	73,588,989	CC	0.127	3.128732		0.261
*mlmm*	LN	AX-108915613	4A	79,319,176	NA	---	3.189661	0.541	0.112
	LN	AX-110600522	6A	111,630,421	CC	0.064	3.099851		0.178
	LN	AX-158544205	7D	24,335,511	CC	0.064	2.900164		0.181
	LN	AX-158612323	3A	3,508,832	NA	---	2.778897		0.233
	LN	AX-109299894	5A	97,047,183	TT	0.288	2.683941		0.163
*Sommer*	LN	AX-158544205	6A	24,335,511	CC	0.064	5.3026811	0.557	0.188
	LN	AX-108741100	6B	67,958,573	NA	---	5.1678802		0.175
	LN	AX-108915613	4A	79,319,176	NA	---	4.6011748		0.110
	LN	AX-109299894	5A	97,047,183	CC	0.108	3.6739810		0.122
	LN	AX-110600522	6A	121,657,421	NA	---	3.5461284		0.135
*NAM*	LN	AX-110385692	3A	47,067,572	TT	0.115	3.541840	0.589	0.288
	LN	AX-158547970	3A	46,623,733	AA	0.112	3.174208		0.276
	LN	AX-158608942	2A	64,279,945	GG	0.224	2.110374		0.234
	LN	AX-86178623	2A	248,384,372	TT	0.213	2.028563		0.260
	LN	wsnp_Ex_c351_689415	7B	706,808,922	CC	0.115	2.016566		0.251

^a^ Chr: Chromosome, ^b^ Pos: Position (bp), ^c^ Maj. Allele: Major Allele, ^d^ MAF: Minor Allele Frequency, ^e^ −log(*p Value*) for five top significant SNPs received from different GWAS models, ^f^ ML $locFDR\left(z_{i}\right)$: maximum likelihood of five significant SNPs based on local FDR threshold in the GWAS model, ^g^ P ($bsFDR\left(\lambda_{SNP}\right)$): posterior probability of SNP when alternative hypothesis is true.

**Table 2 ijms-24-14011-t002:** GWAS results using rrBLUP, mlmm, Sommer and NAM models for GY vector of 221 wheat genotypes under high N treatment.

GWAS Model	N Level	SNP	Chr ^a^	Pos ^b^	Maj. Allele ^c^	MAF ^d^	−log (*p-Value*) ^e^	ML $\left(locFDR\left(z_{i}\right)\right)$ ^f^	*P* $(bsFDR\left(\lambda_{SNP}\right)$) ^g^
*rrBLUP*	HN	AX-158576714	3A	69,099,202	CC	0.181	3.288885	0.561	0.389
	HN	AX-109898892	3A	73,588,989	CC	0.127	3.128732		0.391
	HN	AX-111563200	1B	15,596	TT	0.112	2.764430		0.269
	HN	AX-110038979	5B	13,056	NA	---	2.173367		0.266
	HN	AX-108852922	5B	14,184	NA	---	2.039261		0.266
*mlmm*	HN	AX-109898892	3A	73,588,989	CC	0.127	3.288885	0.454	0.372
	HN	AX-158544205	7D	24,335,511	CC	0.064	2.900164		0.284
	HN	AX-158612323	3A	3,508,832	NA	---	2.778897		0.357
	HN	AX-108915613	4A	79,319,176	NA	---	2.153261		0.251
	HN	AX-109299894	5A	97,047,183	TT	0.288	2.073941		0.272
*Sommer*	HN	AX-108915613	4A	79,319,176	NA	---	4.6011748	0.661	0.248
	HN	AX-109299894	5A	97,047,183	CC	0.108	3.6739810		0.223
	HN	AX-158544205	6A	24,335,511	CC	0.064	3.211453		0.217
	HN	AX-158576714	3A	69,099,202	CC	0.181	3.288885		0.281
	HN	IACX2540	5A	619,684,824	CC	0.213	2.78284		0.272
*NAM*	HN	AX-158576714	3A	69,099,202	CC	0.181	3.288885	0.558	0.415
	HN	AX-109898892	3A	73,588,989	CC	0.127	3.2845915		0.427
	HN	AX-110385692	3A	47,067,572	TT	0.115	3.1852291		0.414
	HN	AX-109470057	3A	61,095,990	CC	0.162	2.5953862		0.427
	HN	BS00094057_51	3A	7,437,103	TT	0.063	1.938022		0.412

^a^ Chr: Chromosome, ^b^ Pos: Position (bp), ^c^ Maj. Allele: Major Allele, ^d^ MAF: Minor Allele Frequency, ^e^ −log(*p* Value) for top five significant SNPs received from different GWAS models, ^f^ ML $locFDR\left(z_{i}\right)$: maximum likelihood of five significant SNPs based on local FDR threshold in the GWAS model, ^g^ P ($bsFDR\left(\lambda_{SNP}\right)$): posterior probability of SNP when alternative hypothesis is true.

## Data Availability

The data that support the findings of this study are available from the corresponding author upon reasonable request.

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
