# Peer review of "Local and Bayesian Survival FDR Estimations to Identify Reliable Associations in Whole Genome of Bread Wheat"

_ijms, 2023, doi:10.3390/ijms241814011_

Round 1

Reviewer 1 Report

The manuscript discusses the challenges of estimating a FDR significance threshold in genome-wide association studies and proposes two methods for identifying reliable associations in the whole genome of bread wheat: the local FDR approach and the Bayesian perspective. The paper highlights the advantages and limitations of each method and provides examples of how they have been used to distinguish true positive hypotheses from false positive and negative errors in genome-wide association studies.

The value of the paper research lies in its contribution to the field of genome-wide association studies by proposing two methods for identifying reliable associations in the whole genome of bread wheat. The paper provides a comprehensive overview of the challenges of estimating a FDR significance threshold and the advantages and limitations of the local FDR approach and the Bayesian perspective.

One drawback of the method used in the paper is that it requires a large sample size to achieve reliable results. Additionally, the paper acknowledges that the local FDR approach may not be suitable for identifying rare variants.

One improvement suggestion could be to explore alternative methods for identifying reliable associations in the whole genome of bread wheat, such as machine learning algorithms or network-based approaches. Additionally, future research could focus on validating the results of the local FDR approach and the Bayesian perspective using independent datasets.

One potential limitation of the paper is that it focuses specifically on the whole genome of bread wheat, which may limit the generalizability of the findings to other crops or organisms. 

Finally, the paper does not provide a comparison of the proposed methods with other existing softwares for identifying reliable associations in genome-wide association studies, which may limit the ability to fully evaluate the effectiveness of the proposed methods.

Further optimization is needed for images, such as unifying font size, improving image resolution, and unifying letter numbering for images. Figure 5 does not have a letter number and should be added.

There are some minor issues in English that need to be addressed and improved. such as line 172, “Six” should not be bold. 

Author Response

Response to reviewers’ comments

We are very grateful for the reviews provided by the editors and each of the reviewers of this manuscript. It was your valuable and insightful comments that led to possible improvements in the current version. The authors have carefully considered the comments and tried our best to address every one of them. We have subjected our manuscript to extensive editing of English language and it has been completely rewritten again. We hope the manuscript after careful revisions meet your high standards. The authors welcome further constructive comments if any. Please, below we provide the point-by-point responses to the comments in blue. All page numbers refers to the manuscript file with tracked changes.

Reviewer 2 Report

The authors in this research article addresses the challenge of estimating a FDR significance threshold in GWAS to differentiate true positive hypotheses from false positive and negative errors. Different multiple testing comparison methods have been developed to determine the significance threshold; however, these methods may be overly conservative and lead to an increase in false negatives. The study discusses two empirical methods to determine the statistical significance threshold based on the SNP-based heritability of grain yield as a complex trait in bread wheat. The local false discovery rate and Bayesian survival FDR approaches are proposed to identify reliable associations in GWAS models. The local FDR approach focuses on SNP effects in the tails of the distribution, potentially missing signal associations distributed across the genome. Bayesian survival analysis is employed to address the challenges of multi-collinearity and large-scale problems within GY vector data. The research aims to obtain a short list of SNPs reliably associated with GY under different nitrogen levels. The study uses field experiments over three years and assesses the GY vector in bread wheat. The top five significant SNPs are compared using various Bayesian models, with Bayesian survival analysis effectively identifying differences between minor and major alleles due to time-to-event considerations. The discussion emphasizes the importance of considering genetic and hyper-parameter effects, such as linkage disequilibrium analysis, minor allele frequency, and call rate, in GWAS models. The limitations of traditional linear GWAS models are highlighted, particularly when dealing with non-additive relationships between loci, such as codominant and epistatic interactions. The study employs various statistical and computational methods, including Bayesian approaches and survival analysis, to address the challenges in GWAS and GP models. The application of these methods to genomic data from bread wheat helps to identify reliable associations and optimize model performance. The article suggests that Bayesian survival analysis is particularly effective in capturing small effects within large-scale genomic data. Overall, the research proposes novel empirical methods for determining significance thresholds in GWAS and demonstrates their effectiveness through the analysis of GY data in bread wheat. The study contributes to the advancement of statistical approaches in genomics research, particularly in addressing the challenges associated with complex traits and large-scale datasets. However, I have a couple of comments/suggestions:

Abbreviate NAM upon its initial mention and provide an explanation.

Occasionally, sentences conclude abruptly, leaving the full meaning unclear.

Ensuring the replicability of the study necessitates the public availability of all phenotypic data and codes. For studies of this nature, establishing a GitHub repository and disclosing all analysis and plotting codes is of utmost importance. This facilitates reanalysis, enables the calculation of proposed multiple corrections using others' datasets, and encourages collaborative efforts. So please make all phenotypes and codes public.

Occasionally, sentences conclude abruptly, leaving the full meaning unclear. There are a lot of grammatical errors and problems associated with sentence structure. The article must be rewritten by a native English speaker.

Author Response

We are very grateful for the reviews provided by the editors and each of the reviewers of this manuscript. It was your valuable and insightful comments that led to possible improvements in the current version. The authors have carefully considered the comments and tried our best to address every one of them. We have subjected our manuscript to extensive editing of English language and it has been completely rewritten again. We hope the manuscript after careful revisions meet your high standards. The authors welcome further constructive comments if any. Please, below we provide the point-by-point responses to the comments in blue. All page numbers refers to the manuscript file with tracked changes.
